# Characteristics of Boundary Layer Turbulence Energy Budget in Shenzhen Area Based on Coherent Wind Lidar Observations

Jinhong Xian[1, 3], Zongxu Qiu[1], Huayan Rao[1], Zhigang Cheng[2], Xiaoling Lin[1], Chao Lu[1], Honglong Yang[1], Ning Zhang[3, 4]

[1] Shenzhen National Climate Observatory, Meteorological Bureau of Shenzhen Municipality, Shenzhen 518040, China
[2] Key Laboratory of Urban Meteorology, China Meteorological Administration, Beijing 100089, China
[3] School of Atmospheric Sciences, Nanjing University, Nanjing 210023, China
[4] Key Laboratory of Urban Meteorology, China Meteorological Administration, Beijing, 100089, China

*Correspondence to*: Honglong Yang (yanghl01@163.com); Ning Zhang (ningzhang@nju.edu.cn)

**Abstract.** Due to the limitations of observations with meteorological towers and aircraft, there is a lack of research on the vertical characteristics of the atmospheric boundary layer in relation to the budget terms of turbulence kinetic energy (TKE). This study reveals the seasonal characteristics of the TKE budget and processes in Shenzhen using long-term observational data from coherent wind lidar. We found that the TKE variations in the region transition in behavior around 14:00 local time, mainly because of changes in buoyancy generation. We determined that TKE is strongest in summer and has the highest impact at high altitudes in autumn in Shenzhen. Our results indicate that above 360 m, the daytime turbulent transport term in all seasons is positive, contributing up to 20% of the total TKE budget, and the dissipation rate term is t is the only factor that dominates energy dissipation. We also found seasonal differences in the vertical characteristics of the dissipation rate in the region, with maximum values observed near the ground during spring, summer, and autumn. Our results indicate that near the ground, buoyancy is the main generation process of TKE, contributing up to 60% of the total budget. Above 570 m, the role of shear generation gradually becomes more prominent, comparable to buoyancy generation. These findings not only enrich our understanding of the vertical structure of atmospheric turbulence, but also provide new observational data and theoretical support for the parameterization of the turbulence energy budget in climate models, which can help improve atmospheric predictions.

## 1 Introduction

The vertical characteristics of atmospheric turbulence kinetic energy (TKE) are of great significance for understanding energy conversion, material exchange, and the evolution of weather systems in the atmosphere (Stull, 1988; Kaimal et al., 1976; Heilman et al., 2018). By delving into these characteristics, we can better understand the dynamic behavior and energy balance mechanism of the atmospheric system, providing strong support for meteorological observations, forecasting, air quality control, and disaster prevention and reduction (Stull, 1988; Wyngaard, 2010; Caughey and Wyngaard, 1979; Song, 2021).

Previously, research on the atmospheric TKE budget mainly relied on near-ground meteorological tower or aircraft observations, as well as advanced numerical simulations and parameterization schemes (Deardorff, 1974; Endoh et al., 2014; Puhales et al., 2013; Therry and Lacarrere, 1983; Zeman and

Tennekes, 1977; Zhou et al., 1985; Elguernaoui et al., 2023; Nilsson et al., 2016b; Canut et al., 2016; Li et al., 2023; Guo et al., 2021; Meng et al., 2024; Yus-Diez et al., 2019). Lenschow (1974) found, based on aircraft measurements and surface layer observations, that turbulent transport increases almost linearly with altitude, balancing the almost linear decrease in buoyancy generation. In addition, their results showed that under the neutral stability limit, the transport term in the budget equation is the smallest, and the shear generation term can be ignored at about 10 times the Obukhov length above the ground. Chou et al. (1986) combined vertical lidar and aircraft observations to reveal the role of turbulent transport in vortex regions, particularly near the top of the mixing layer. Frenzen and Vogel (1992) found through ground observations that in the neutral surface layer, the dissipation rate is 15% to 20% lower than the energy generated by shear generation. Darbieu et al. (2015) studied the attenuation of boundary layer turbulence during the afternoon transition period using dense observational data from field experiments and large eddy simulation (LES) data, with a particular focus on changes in the vertical structure of turbulence. They pointed out that shear generation has a significant impact on the attenuation of TKE during the afternoon transition period, especially at the top of the mixing layer. Nilsson et al. (2016a) analyzed in detail the various components of the TKE budget, including a tendency term, buoyancy generation term, dissipation term, and transport term, through near-surface data obtained with small towers, revealing the differences in surface layer dynamics of TKE and its attenuation during different afternoon periods. Jensen et al. (2017) used data collected during a multiyear wind resource assessment study conducted in a multi-land-use environment along the coast of Belize, and found that in coastal areas, shear generation is more important than buoyancy generation in the TKE budget. Barman et al. (2019) conducted data analysis using a three-dimensional fast response acoustic anemometer at heights of 6 m, 18 m, and 30 m, combined with field observations and numerical simulations. They found that shear generation contributed to the TKE in the afternoon, working together with buoyancy generation. Similarly, Pozzobon et al. (2023) collected turbulence observational data from four different height layers over a period of 10 months based on observational data from a 30 m high meteorological tower. They conducted an in-depth analysis of the TKE budget under daytime convective conditions and nighttime stable conditions, and found that the TKE budget during the daytime was mainly dominated by shear generation and buoyancy generation, while at night it was mainly dominated by shear generation and dissipation rate.

From these existing research results, it can be seen that there are still many challenges in studying the vertical characteristics of the many TKE budget terms. For example, current research methods mostly rely on LESs or data from several atmospheric layers measured by meteorological towers, aircraft, and/or sounding equipment, all of which have limited data continuity and detection height. Although wind profile radar can provide valuable turbulence data, its spatial and temporal resolution might be insufficient, and its ability to monitor turbulence under clear sky conditions is limited (Solanki et al., 2022). In addition, the research period is usually during the daytime or afternoon transition period, usually given in the form of a profile, meaning continuous vertical spatial feature distributions are lacking. Moreover, there are relatively few research works based on a single detection method.

How to distinguish and quantify TKE budget terms at different heights accurately, and how to integrate these observational data into climate models effectively, are currently hot research topics.

Future research needs to combine more continuous vertical observational data to gain a deeper understanding of the vertical characteristics of atmospheric turbulence and its impact on weather and climate change. In our previous research, we directly acquired atmospheric turbulence parameters from the perspective of spectral analysis using coherent wind lidar (Xian et al., 2024b; Xian et al., 2024c). On this basis, we here propose a detection method for TKE budget terms based on coherent wind lidar and strict data quality control, to ensure the accuracy and reliability of the measured data (Xian et al., 2025). By comparing with data obtained from three-dimensional ultrasonic anemometers, we show that the average absolute error is less than 0.00014 $m^2/s^3$, which verifies the accuracy and reliability of our method. The proposed method can provide the spatiotemporal distribution characteristics of parameters such as TKE, dissipation rate, shear generation, turbulent transport, and buoyancy generation. Based on this method, in this study we analyze the vertical structural characteristics and variation laws of shear generation, buoyancy generation, turbulent transport, and dissipation rate in the Shenzhen area.

The rest of the paper is organized as follows. In Section 2, we introduce the equipment and data quality control methods used, as well as the methods for obtaining the various TKE budget terms based on wind lidar. In Section 3, we conduct statistical analysis on the spatiotemporal variation characteristics of the various turbulent TKE budget terms and explore the variational patterns of each TKE budget term in different seasons. The main conclusions of this study are presented in Section 4.

## 2 Instrument, Data, and Methods

Shenzhen is located south of the Tropic of Cancer (114°E, 22.5°N), with a subtropical maritime climate characterized by warm and humid weather, and abundant rainfall. According to the geographical location of Shenzhen, March, April, and May are spring; June, July, and August are summer; September, October, and November are autumn; and December and January and February are winter (Lang et al., 2007).

The Shiyan Observation Base (113.90586°E, 22.65562°N) is located on the outskirts of Shenzhen. Its unique geographical location makes it an ideal place for conducting meteorological research. The surrounding environment of the base has not been disturbed by large artificial structures, with agricultural land located 1–2 km northeast, while the terrain in the south and northwest is relatively flat, mainly composed of forests and lakes. The base has the tallest meteorological gradient observation tower in Asia, with a height of 356 m. At the bottom of the gradient observation tower, a coherent wind lidar is deployed, as shown in Figure 1. Given that there are no obstacles around the base, the collected data have important reference value in the field of meteorology (Zhou et al., 2023). According to the configuration information provided in Table 1, the parameter configuration of the wind lidar (DSL-W, Darsunlaser Technology Co., Ltd., Shenzhen, China) has been recorded in detail. In previous studies, our research team comprehensively validated the accuracy of the lidar data, ensuring its reliability in subsequent scientific research (Xian et al., 2024b).

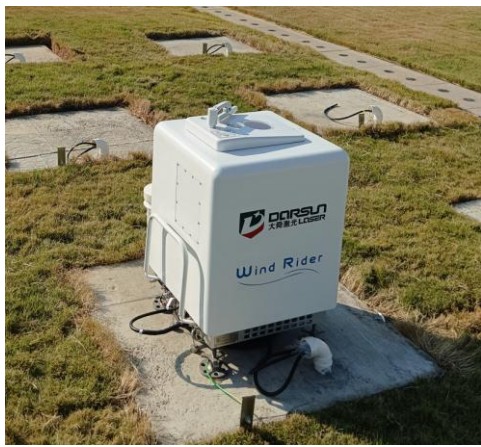

**Figure 1.** Installation diagram of the wind lidar.

**Table 1.** Performance parameters of the wind lidar instrument

| | Metrics | Technical Performance Requirements |
|---|---|---|
| Wind Lidar | Minimum detection altitude | $\leqslant$30 m |
| | Maximum detection altitude | 3 km |
| | Distance resolution | 30 m |
| | Temporal resolution of wind profile | 0.2Hz / 5 s |
| | Errors of wind speed | $\leqslant$0.3 m s$^{-1}$ |
| | Errors of wind direction | $\leqslant$3° |
| | Range of wind speed measurement | 0–60 m s$^{-1}$ |
| | Range of wind direction measurement | 0°– 360° |

To ensure the reliability of the observational data, we performed quality control on the TKE budget term data estimated by the wind lidar. For each height level, we calculated the standard deviation of the data every 30 minutes and eliminated data points that deviated from the mean by more than three standard

deviations. This is based on the commonly used "triple standard deviation principle" in statistics, which means that the probability of data points falling outside the mean plus or minus three standard deviations is very small (close to 0); any points that are greater than three standard deviations away from the mean are considered outliers. We repeated this process three times to ensure accurate identification and handling of outliers. For a single profile dataset, if more than 20% of the data points below 500 m were

lost, the entire profile was discarded, and discontinuous or missing values were estimated using linear interpolation. If the number of lost measurements exceeded 20% within 30 minutes, we discarded the data for that period. Overall, this data quality control process aims to ensure the accuracy, completeness, and reliability of the data through statistical methods.

The wind lidar has 355/365 days of available data throughout 2022, for an availability rate of 97%.

To ensure universality and generalizability of the results, we analyzed the turbulent characteristics of only sunny days and days without low clouds—resulting in data from 279 days. The number of data samples per month is shown in Table 2. Of note, the missing observation days in June and August are due to adverse weather conditions, including continuous cloud cover and heavy precipitation, rather than

data collection issues. From this, it can be seen that the monitoring data of the wind lidar are relatively continuous, and the samples have high representativeness.

**Table 2.** Number of data samples per month in 2022

| Month | Available Days |
|---|---|
| January (Winter) | 31 |
| February (Winter) | 28 |
| March (Spring) | 23 |
| April (Spring) | 27 |
| May (Spring) | 15 |
| June (Summer) | 11 |
| July (Summer) | 27 |
| August (Summer) | 12 |
| September (Autumn) | 29 |
| October (Autumn) | 31 |
| November (Autumn) | 14 |
| December (Winter) | 31 |

The TKE budget equation can be expressed as follows (Stull, 1988; Nilsson et al., 2016a):

$$\frac{\partial E}{\partial t} = \underbrace{-\overline{u'w'}\frac{\partial u}{\partial z} - \overline{v'w'}\frac{\partial v}{\partial z}}_{S} + \underbrace{\frac{g}{\theta_m}\overline{w'\theta'}}_{B} - \underbrace{\frac{\partial \overline{w'E'}}{\partial z}}_{T_t} - \underbrace{\frac{\partial \overline{w'p'/\rho_0}}{\partial z}}_{T_p} - \underbrace{\varepsilon}_{D} , \tag{1}$$

where $E$ represents the TKE (m$^2$/s$^2$), $t$ is the time (s), and $u'$ (longitudinal direction), $v'$ (latitudinal direction), and $w'$ (vertical direction) are the fluctuation values of the three-dimensional wind speed components $u$, $v$, and $w$, respectively, which vary with height above ground level $z$. $g$ is gravitational acceleration, $\theta_m$ is the average potential temperature, $\theta'$ is the fluctuation value of the potential temperature $\theta$, $\rho_0$ is the air density, $P'$ is the fluctuation value of the air pressure $P$, and $\varepsilon$ is the average dissipation rate of TKE. On the left side of the equation is the tendency term ($Et$), while on the right side are the budget terms for shear generation ($S$), dissipation rate ($D$), turbulent transport ($T_t$), pressure transport ($T_P$), and buoyancy generation ($B$). Based on our previous detection method, assuming $T_P$ is ignored (Kaimal and Finnigan, 1994; Wyngaard, 2010; Pozzobon et al., 2023), and obtaining dissipation rate through turbulence inertial subrange spectrum fitting, we obtained the horizontal wind speed (a), vertical wind speed (b), TKE (c), $Et$ (d), $T_t$ (e), $D$ (f), $S$ (g), and $B$ (h) with a time resolution of 20 minutes and a spatial resolution of 30 m (Xian et al., 2025), as shown in the various panels in Figure 2. Because the time resolution was 20 minutes, ensuring steady turbulence within this period was essential. Therefore, we strictly applied a stationarity test, wherein the average variance over the entire time period was required to be similar to the average variance over shorter time intervals (Xian et al., 2025). Using this method, we obtained a dataset of TKE budget terms for the entire year of 2022 in the Shenzhen area. Based on the quality control process, we organized the dataset and analyzed its vertical characteristics in different seasons.

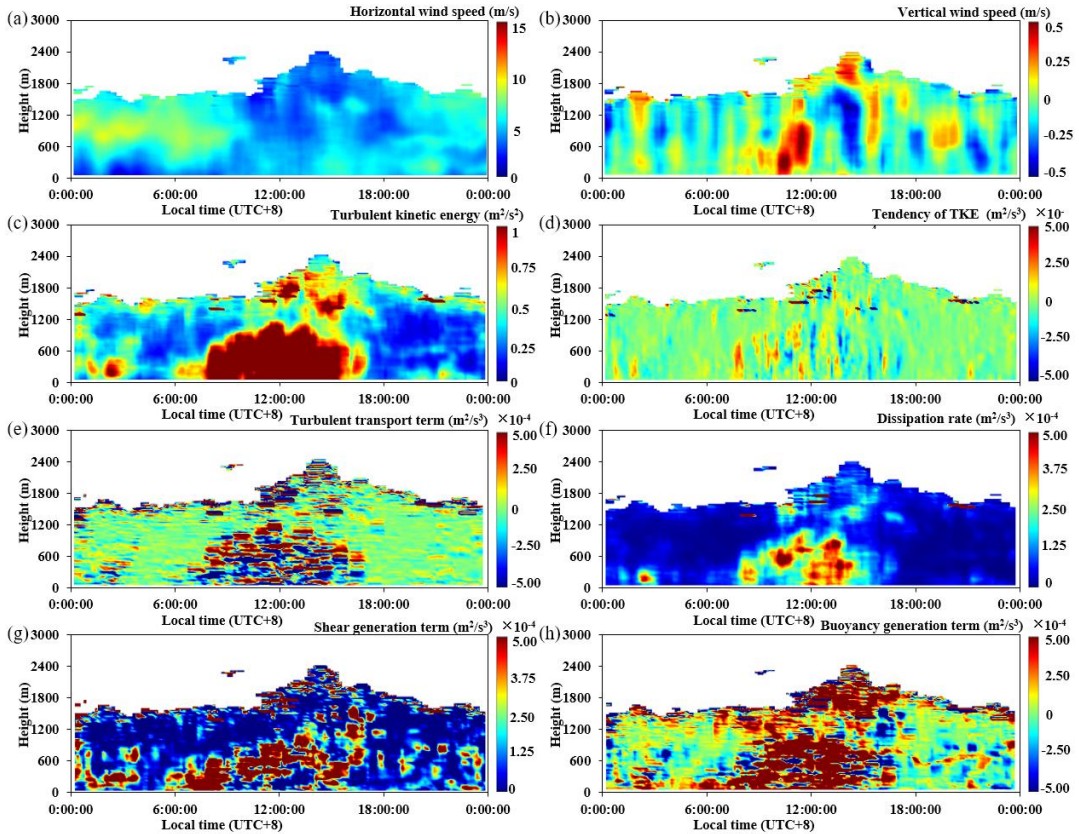

**Figure 2.** Temporal and spatial distributions of the horizontal wind speed (a), vertical wind speed (b), TKE (c), tendency term (d), turbulent transport term (e), dissipation rate (f), shear generation term (g), and buoyancy generation term (h) on September 21, 2022.

## 3 Results and Discussion

### 3.1 Characteristics of the Turbulence Kinetic Energy and Tendency Term

Figure 3 shows the spatiotemporal distribution of the average 24-hour TKE for each season in 2022. It can be seen that the TKE in each season is usually smaller at night and in the morning, and larger during the day. In addition, the TKE also exhibits an uneven vertical distribution, decreasing with increasing height due to the weakening of ground friction and heat supply with increasing height. Furthermore, we present temporal variation curves of the TKE at a height of 120 m for the different seasons, as shown in Figure 4(a). From the graph, it can be seen that the TKE in summer remains the strongest, reaching 1.7 $m^2/s^2$. The maximum TKE values in spring, autumn, and winter are relatively close, around 1.2 $m^2/s^2$. Figure 4(b) shows mean TKE profiles at 13:00 for each season (all times mentioned in this study are in local time). From the figure, it can be seen above 600 m, the TKE in autumn is the strongest, while the turbulence in summer is strongest below 600 m, but rapidly decays above 600 m.

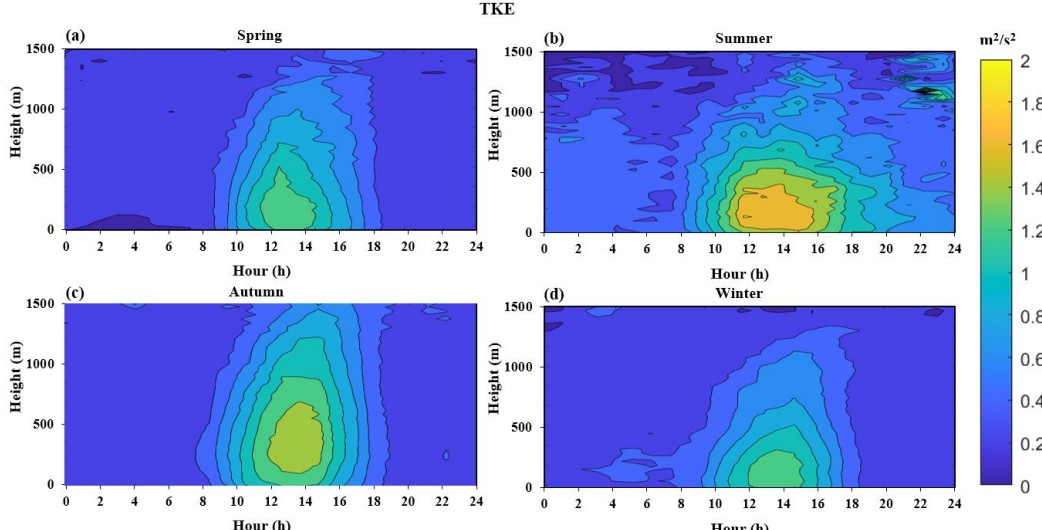

**Figure 3.** Twenty-four hour spatiotemporal distribution map of the TKE in different seasons: (a) spring, (b) summer, (c) autumn, and (d) winter.

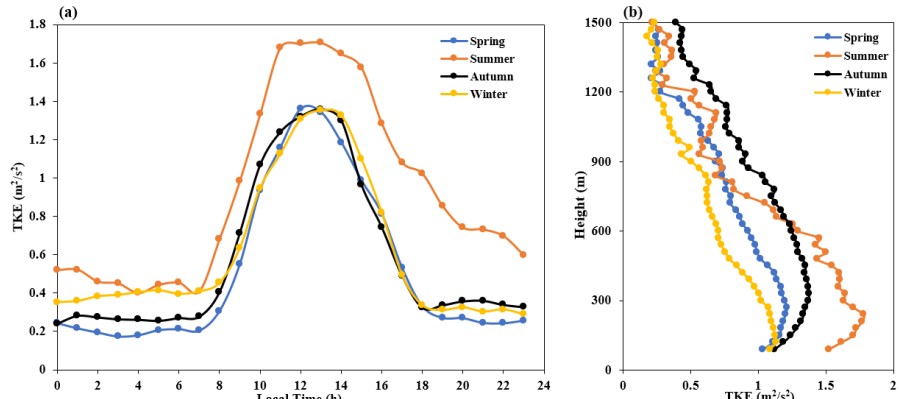

**Figure 4.** Temporal variation of the TKE at a height of 120 m in different seasons (a), and the distribution profile with height at 13:00 for each season (b).

By taking the time derivative of distributions shown in Figure 3, we can obtain the 24-hour spatiotemporal distribution of the TKE tendency terms for the four seasons of 2022, as shown in Figure 5. From this, it can be seen that during the period from 20:00 to 07:00 every day, the tendency term is close to zero, indicating that turbulence tends to stabilize at night. From 08:00 to 14:00 every day, the tendency term of the TKE is greater than zero, indicating an increase in the TKE. From 15:00 to 19:00, the tendency term of the TKE is less than zero, indicating a decrease in the TKE.

We extracted data at a height of 120 m and obtained time variation curves and the corresponding extreme value changes of the TKE tendency terms in the different seasons at a height of 120 m, as shown in Figure 6. From Figure 6(a), it can be seen that the strongest enhancement effect on TKE is at 10:00, and the strongest weakening effect is at 15:00 or 16:00. The peak of the turbulence tendency term is highest in summer, followed by autumn and spring, and is weakest in winter. Similarly, the turbulence tendency term in autumn is the largest of the four seasons. This corresponds to the seasonal characteristics seen in the behavior of the TKE mentioned earlier.

Next, we extracted and summarized the maximum and minimum values of the tendency term for each season, which are plotted in Figure 6(b). We can observe that the TKE tendency term transitions

around 14:00 for each season. To investigate which TKE budget term causes this change, we further differentiated the buoyancy generation terms, $B$ and $S$, at a height of 120 m over time, and obtained their rates of change, $B'$ and $S'$, as shown in Figures 6(c) and (d). From the graphs, we can see that the tendency term of $B$ exhibits the same symmetry as found for the TKE tendency term, while the shear generation term has no obvious pattern, indicating that the main reason for the symmetry of the turbulence tendency term is due to the buoyancy generation term, $B$. In order to explain these seasonal characteristics of the TKE and tendency terms, the seasonal features of the buoyancy generation term, shear generation term, turbulent transport term, and dissipation rate were further analyzed.

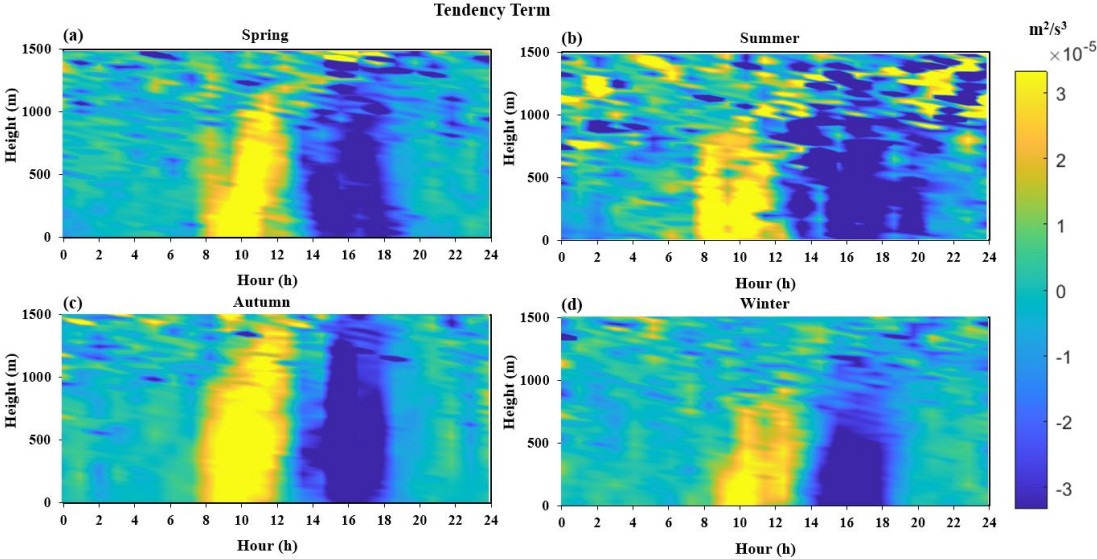

**Figure 5.** Twenty-four hour spatiotemporal distribution map of the TKE tendency term in different seasons: (a) spring, (b) summer, (c) autumn, and (d) winter.

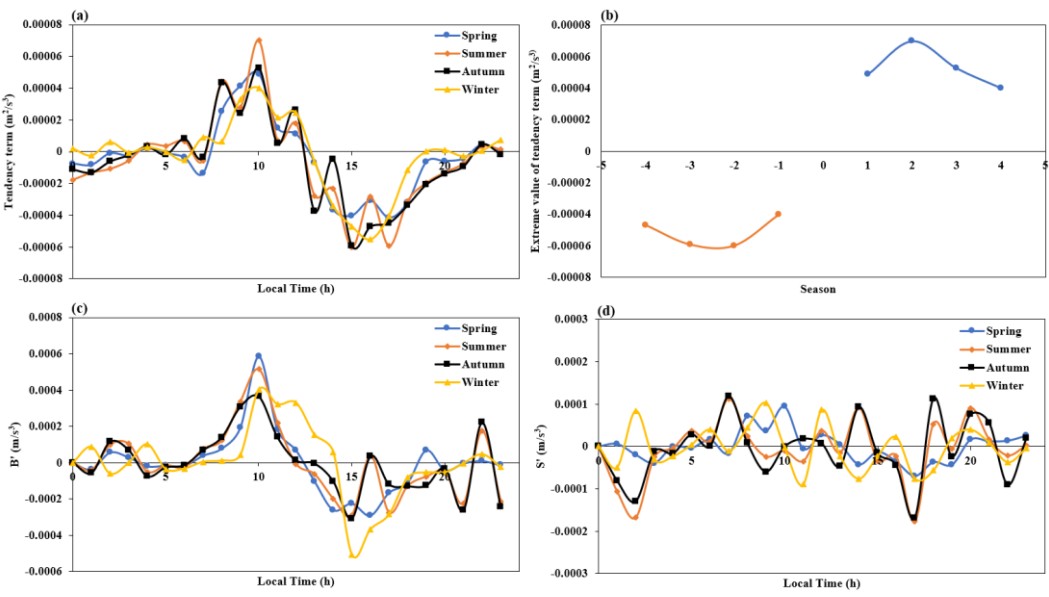

**Figure 6.** Time variation curve (a) and corresponding extreme value variations (b) of the TKE tendency term in different seasons at a height of 120 m, as well as the corresponding change rate of the buoyancy generation term (c) and shear generation term (d).

## 3.2 Characteristics of the Buoyancy Generation Term

Figure 7 shows the spatiotemporal distribution of the 24-hour buoyancy generation term for the four seasons in 2022. It can be seen that the buoyancy generation term shows a relatively obvious daily variation pattern, with smaller values at night and in the morning (generally negative or close to zero) and larger values during the day. Additionally, it can be observed that the impact of the height of the buoyancy generation varies with the seasons, with the highest in autumn and the lowest in winter. Among all seasons, the longest duration of positive buoyancy generation during the daytime is in summer (from 08:00 to 22:00). This indicates that during this period, turbulence is generated and develops, as measured with the buoyancy generation term. This indicates that when the buoyancy generation term in summer is used as the turbulence generation term, its effect time is longer. Figure 8(a) shows time-varying curves of the buoyancy generation term at a height of 120 m in the different seasons. From this, it can be seen that there is not much difference in the values of the buoyancy generation term during the daytime in the different seasons, indicating that the strongest TKE in summer is not due to buoyancy generation. Furthermore, we provide buoyancy generation term for the different seasons and their distribution profiles with height at 13:00 local time, as shown in Figure 8(b). In this figure, it can be seen that the buoyancy generation term is strongest in autumn above 600 m, which provides a reason why the TKE has the greatest impact on height in autumn.

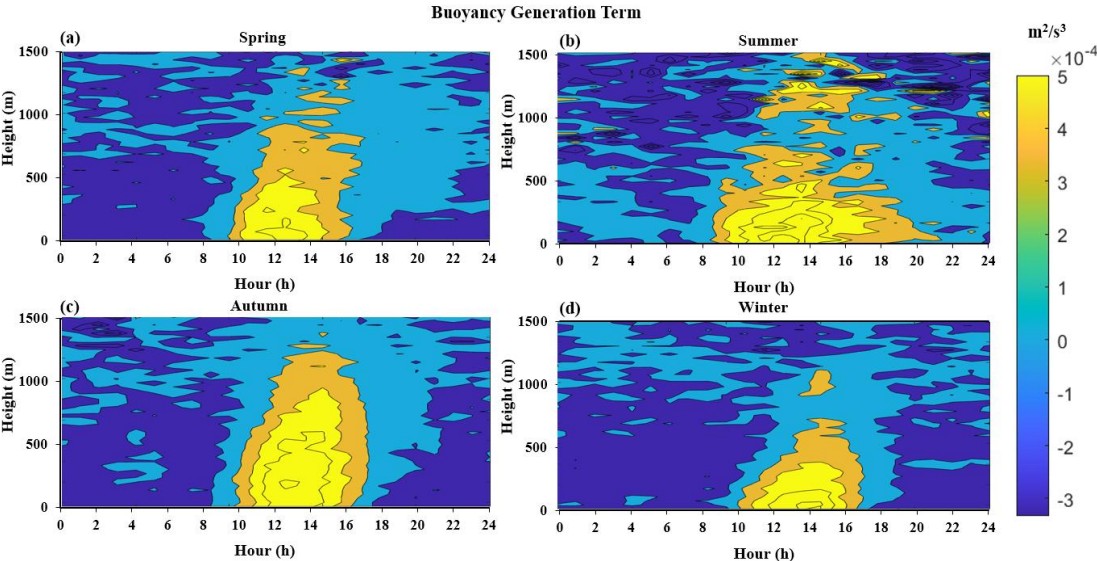

**Figure 7.** Twenty-four hour spatiotemporal distribution of the buoyancy generation term in different seasons: (a) spring, (b) summer, (c) autumn, and (d) winter.

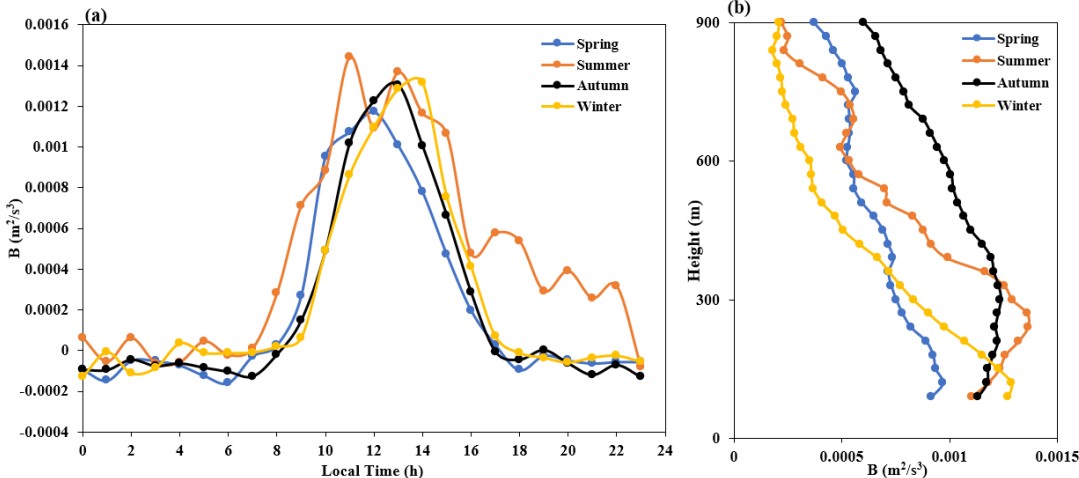

**Figure 8.** Time-varying curves of the buoyancy generation term in the different seasons at a height of 120 m (a), as well as the distribution profile with height at 13:00 local time for each season (b).

### 3.3 Characteristics of the Shear Generation Term

Figure 9 shows the spatiotemporal distribution of the shear generation term over 24 hours for the four seasons in 2022, which is always positive throughout the day. In terms of magnitude, it usually reaches a maximum during lunchtime. In addition, the shear generation term has significant nighttime behavior in summer. Figure 10 shows the spatiotemporal distribution of the horizontal wind speed over 24 hours in the different seasons of 2022. From this, it can be seen that the nighttime low-level jet is more pronounced in summer than in the other seasons, which is consistent with our previous research on the characteristics of the low-level jet in the Pearl River Delta (Qiu et al., 2023). Therefore, we can conclude that the shear generation term in summer is significant during the nighttime due to the climate characteristics of the region, which produce more low-level jets in summer. Figure 11(a) shows time-varying curves of the shear generation term at a height of 120 m for the different seasons in 2022. From this, it can be seen that the shear generation term in summer is larger than in other seasons, similar to the behavior seen in Figure 4(a). Furthermore, we have plotted the shear generation term for the different seasons and their distribution profiles with height at 13:00 local time, as shown in Figure 11(b). It can be seen that below 600 m, the shear generation term is highest in summer, corresponding to the strongest TKE below 600 m in summer (Figure 4(b)). From this, we can see that we have identified the reason for the strongest turbulence in summer, which is caused by the generation of shear. In addition, above 600 m, the shear generation term is strongest in autumn. From this, we can see that another reason why TKE has the greatest impact at height in autumn is shear generation.

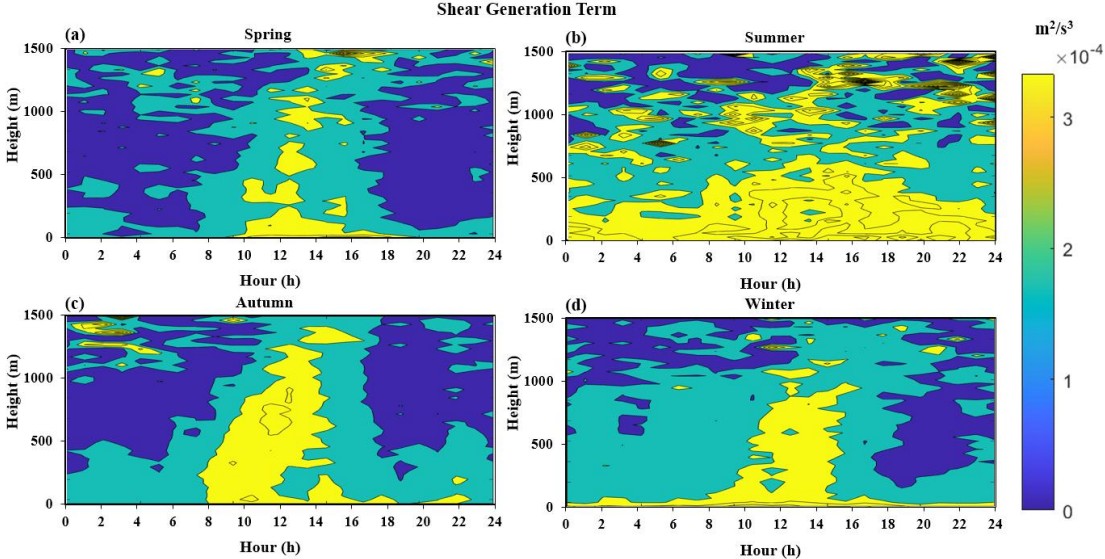

**Figure 9.** Twenty-four hour spatiotemporal distribution map of the shear generation term in the different seasons: (a) spring, (b) summer, (c) autumn, and (d) winter.

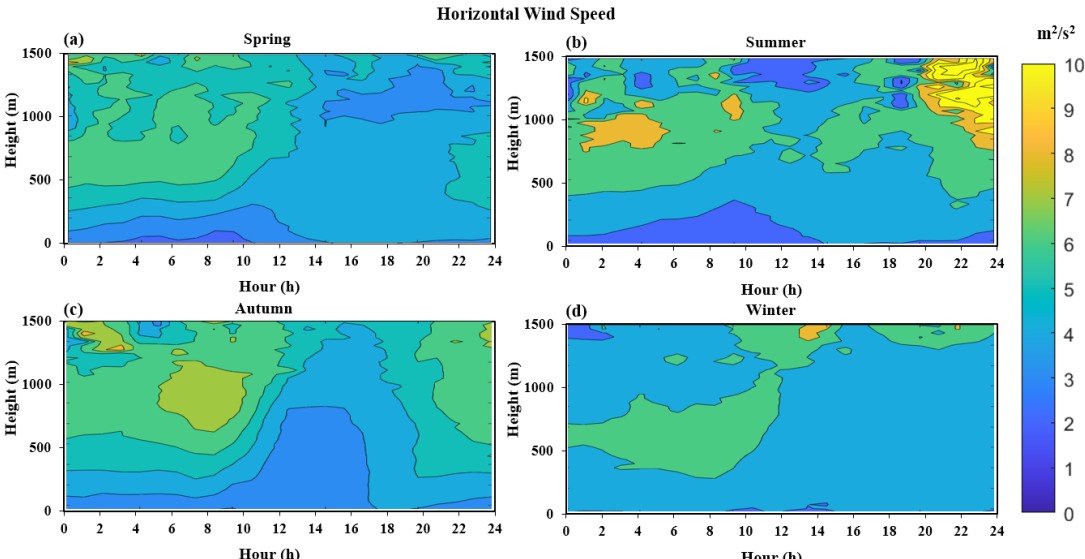

**Figure 10.** Twenty-four hour spatiotemporal distribution map of the horizontal wind speed in the different seasons: (a) spring, (b) summer, (c) autumn, and (d) winter.

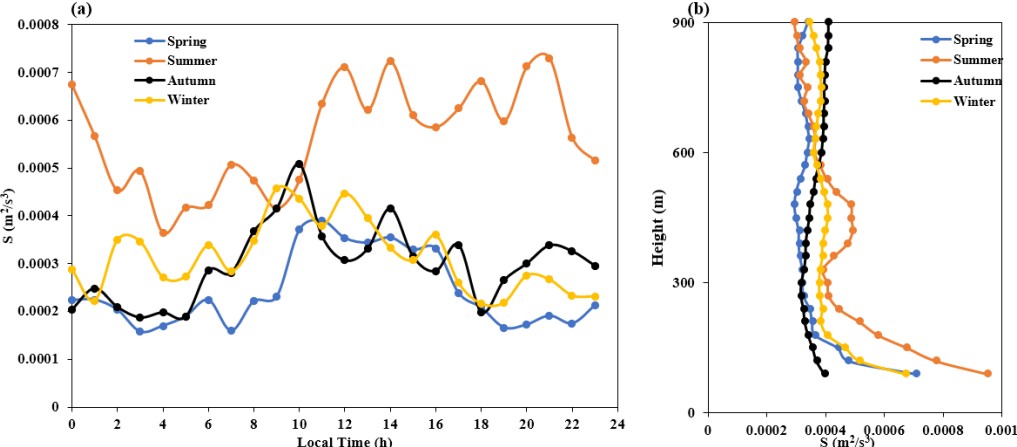

**Figure 11.** Time-varying curves of the shear generation term in the different seasons at a height of 120 m (a), as well as the distribution profile with height at 13:00 local time (b).

## 3.4 Characteristics of the Turbulent Transport Term

Figure 12 shows the 24-hour spatiotemporal distribution of the turbulent transport term for the four seasons in 2022. Unlike the characteristics of the TKE, buoyancy, and shear generation terms mentioned earlier, the turbulent transport term exhibits a negative value near the ground throughout the day and a zero or positive value at high altitudes. We note that previous research based on tower base observations have shown that the turbulent transport term is negative near the ground (Nilsson et al., 2016a). However,

due to limitations in current observation methods, previous studies were unable to observe the turbulent transport term at high altitudes directly. From the results presented here, not only can we intuitively see that the turbulent transport term at high altitudes is positive in the different seasons, but we can also intuitively see its spatiotemporal distribution.

      Figure 13(a) shows temporal variation curves of the turbulent transport term at a height of 120 m in

the different seasons. From this, it can be seen that turbulent transport has the smallest effect in autumn and the largest effect in summer. Figure 13(b) shows height distribution profiles of the turbulent transport term in the different seasons at each day at 13:00 local time. From this, it can be seen that above 360 m, the turbulent transport term in all seasons is positive, and above 780 m, it tends to be consistent. At around 600 m, it can be seen that the turbulent transport term in summer begins to decrease, which is

similar to the rapid decay of the TKE curve around 600 m in summer. This indicates that above 600 m, the inflow of turbulent energy begins to decrease, resulting in a decrease in the TKE.

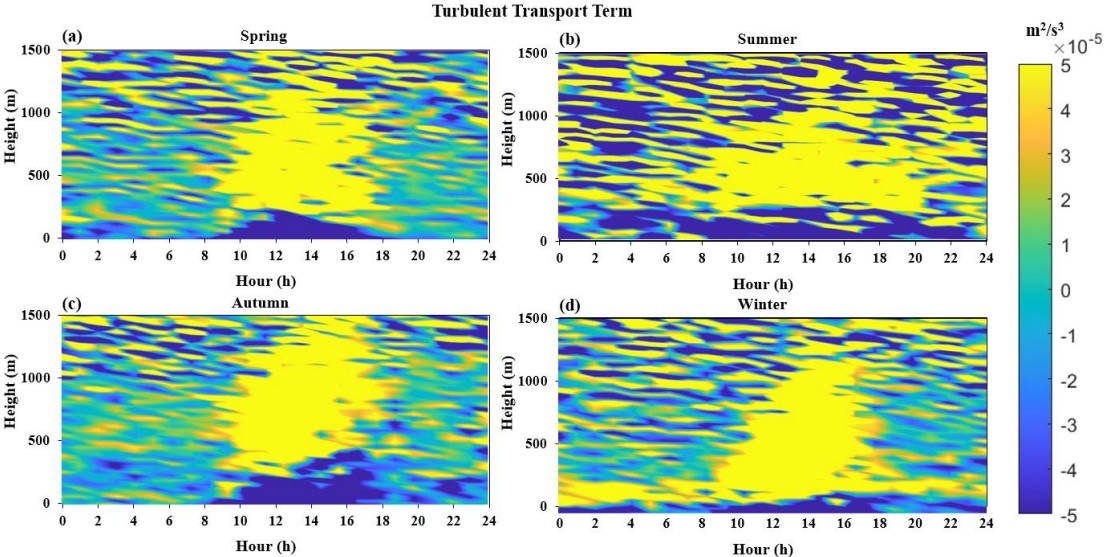

**Figure 12.** Twenty-four hour spatiotemporal distribution map of the turbulent transport term in the different seasons: (a) spring, (b) summer, (c) autumn, and (d) winter.

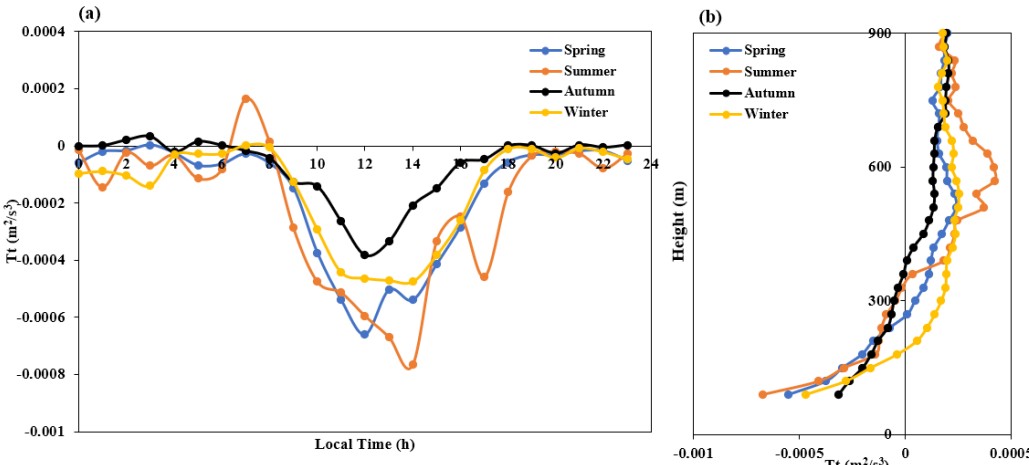

**Figure 13.** Time-varying curves of the turbulent transport term in the different seasons at a height of 120 m (a), as well as the distribution profile with height at 13:00 for each season (b).

Figure 14 shows variation curves of the spatial derivatives of the turbulent transport term, buoyancy generation term, and shear generation term, as well as a comparison and the corresponding correlations in their continuous changes over one year at a height of 120 m. From this, it can be seen that there is a significant correlation ($R > 0.76$) between the rate of change of the buoyancy and shear generation terms in space and the turbulent transport term.

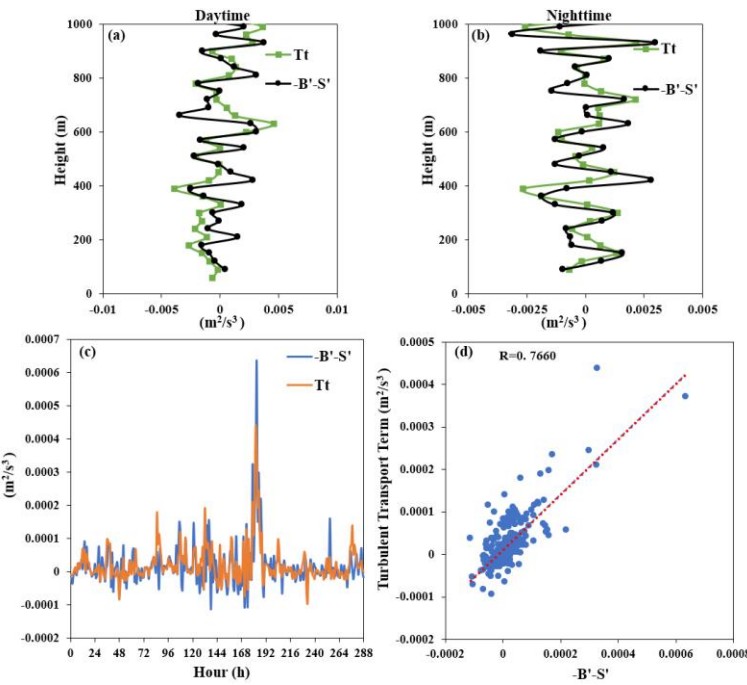

**Figure 14.** Variation curves of the spatial derivatives of the turbulent transport term, buoyancy generation term, and shear generation term during the daytime (a), at night (b), at a height of 120 m over the whole year (c), and the corresponding correlation (d).

### 3.5 Characteristics of the Dissipation Rate

Figure 15 shows the spatiotemporal distribution of the 24-hour dissipation rate for the four seasons of 2022. It can be seen that during the day, due to solar radiation heating the surface, an unstable boundary

layer forms, which contains strong turbulent motion and a high dissipation rate. At night, the surface is cooled by the ground emitting infrared radiation, which leads to the formation of a stable boundary layer with weak turbulence and a low dissipation rate. Similar to the shear generation that has a significant nighttime effect in summer, the dissipation rate also exhibits a relatively large characteristic in summer. In order to clarify the factors that cause this large dissipation rate, we calculated the correlation between it and the shear generation and buoyancy generation terms, as shown in Figure 16. From this plot, it can be seen that the dissipation rate term is significantly correlated with the shear generation term at night ($R > 0.9$), but not with the buoyancy generation term ($R < 0.4$). Thus, we determine that the increase in the nighttime dissipation rate is caused by shear generation.

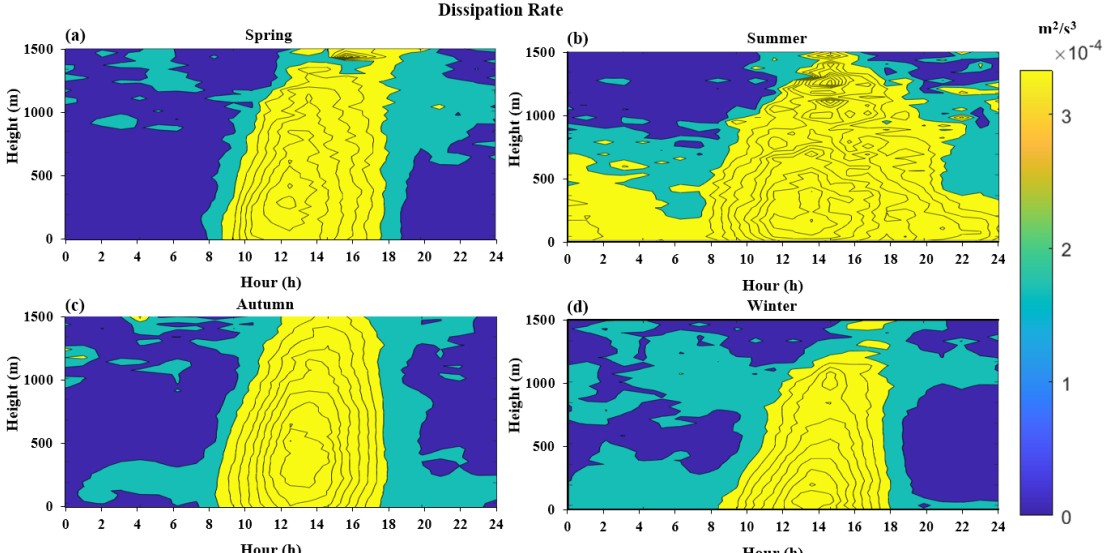

**Figure 15.** Twenty-four-hour spatiotemporal distribution of the dissipation rate term in the different seasons: (a) spring, (b) summer, (c) autumn, and (d) winter.

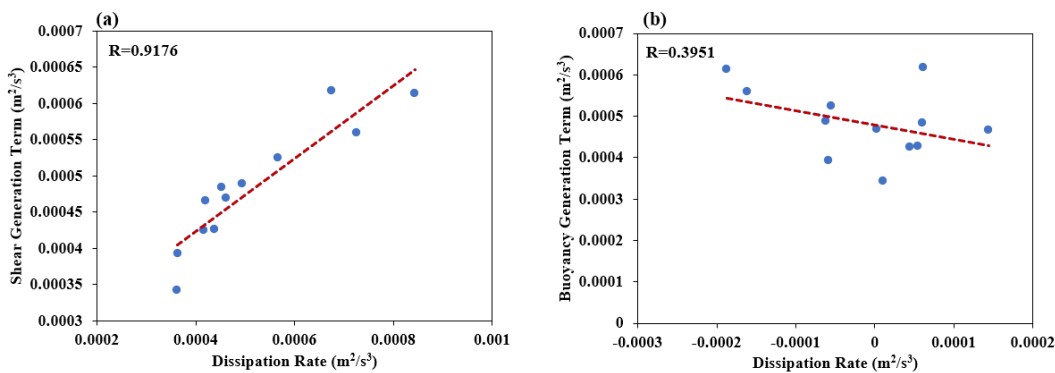

**Figure 16.** Correlation between the dissipation rate and the shear generation term (a) and buoyancy generation term (b) during the night.

Figure 17(a) shows time-varying curves of the dissipation rate at a height of 120 m for the four seasons in 2022. From this, we find that the dissipation rate in summer is higher than in the other seasons, while the dissipation rate in winter is higher than that in spring and autumn, and not significantly different from that in summer. Figure 17(b) shows distribution profiles of the dissipation rate with height for the four seasons in 2022 at 13:00 each day. From this, it can be seen that the dissipation rate is strongest above 600 m, and it has the greatest effect in autumn. Correspondingly, in Figure 4(b), the TKE is

strongest in autumn above 600 m, indicating that the dissipation rate is proportional to the TKE at high
altitudes. Considering the positive value of the turbulent transport term above 360 m, it can be inferred
that the dissipation rate term is the only factor that dominates energy dissipation at heights above 360 m
during the daytime. Below 360 m, TKE is not directly proportional to the dissipation rate because the
transport of turbulence acts as a form of dissipation. In addition, it can be observed from the graph that
the dissipation rate reaches a maximum near the ground, which is different from what was found in
previous studies and deserves further investigation (Zhou et al., 1985).

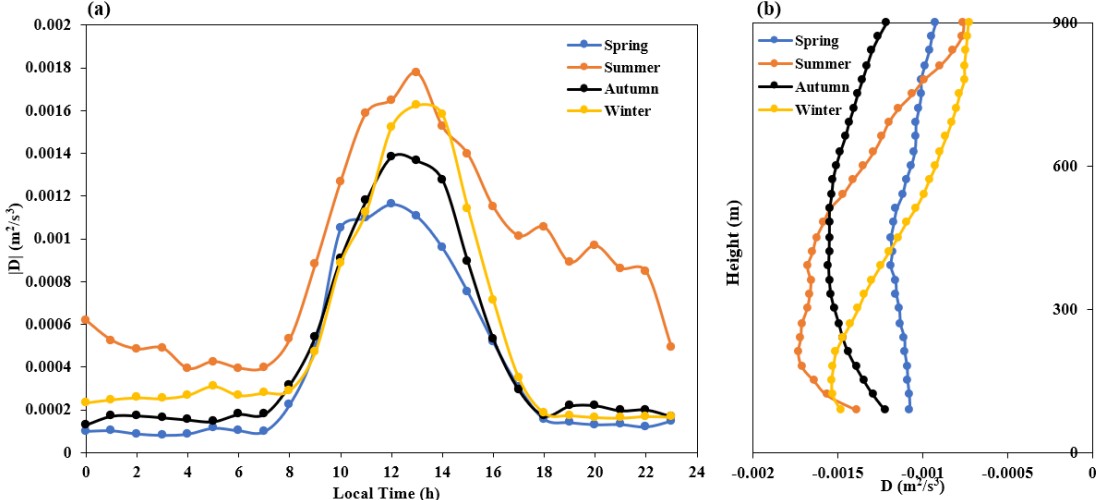

**Figure 17.** Time-varying curves of the dissipation rate in the different seasons at a height of 120 m (a), as well as
the distribution profile as a function of height at 13:00 local time (b).

Figure 18 shows the daytime changes of the TKE budget terms at different heights in the boundary
layer for the different seasons. We first categorized the TKE budget terms into two groups according to
positive (including zero) and negative values, which were designated as generation and dissipation terms,
respectively. Subsequently, we calculated the contribution rate of each budget term within its
corresponding category by determining its proportion. From this, it can be seen that below 600 m in the
four seasons, the buoyancy generation term is the most important, contributing up to 60% of the entire
TKE budget. In summer and winter, above 570 m, the contribution of the shear generation term is
equivalent to that of the buoyancy term. In spring, summer, and autumn, turbulent transport plays a very
important role at altitudes of about 360 m, contributing up to 20% and being an energy inflow process
that cannot be ignored. This result has important reference significance for improving parameterization
schemes of the boundary layer. In the previous section, we analyzed the buoyancy and shear generation
terms and concluded that they are the reasons for the highest impact height of TKE in autumn. From
Figure 18(c), we can further see that the contribution of the buoyancy generation term reaches 60%,
which is much larger than the 20% contribution of the shear generation term. Therefore, we can conclude
that the reason for the high-altitude impact of TKE in autumn is buoyancy generation.

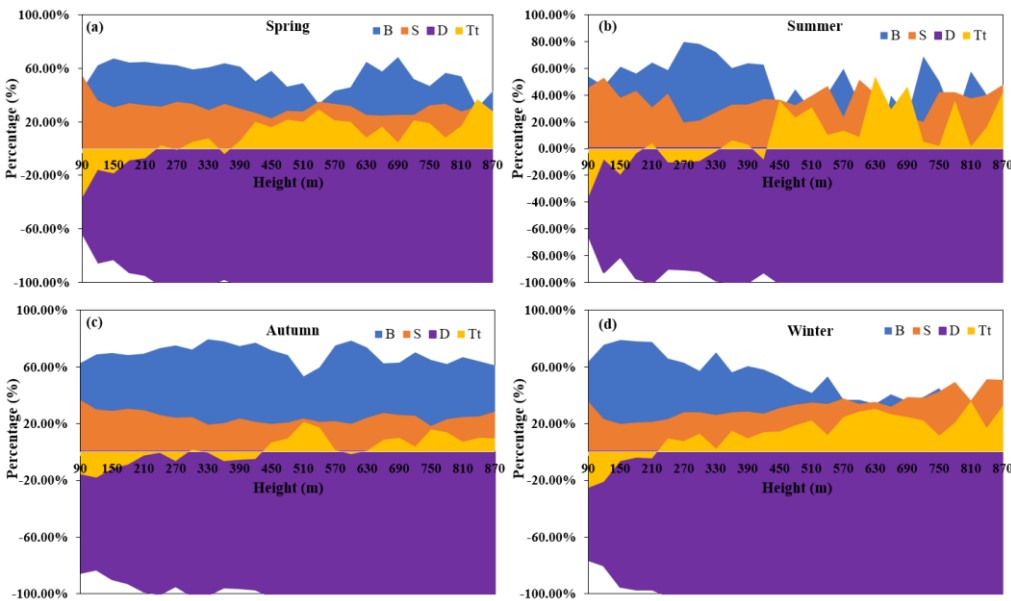

 **Figure 18.** Proportions of buoyancy generation, shear generation, turbulent transport, and dissipation rate at different heights in the boundary layer during the day for each season: Spring (a), Summer (b), Autumn (c), and Winter (d).

## Conclusion

In this study, we used wind lidar to estimate atmospheric turbulence energy to reveal the seasonal characteristics of the turbulent energy process, and its various components, in the boundary layer over the Shenzhen area. We revealed that the TKE tendency term in the Shenzhen area transitions around 14:00, which is mainly caused by buoyancy generation. At the same time, we found that the TKE in the Shenzhen area is strongest in summer and has the highest height in autumn. By analyzing the characteristics of the buoyancy and shear generation terms, we revealed that the strongest turbulence in Shenzhen during summer is caused by shear generation, while the largest TKE component in autumn is buoyancy generation.

Regarding the turbulent transport term, it is positive during the daytime at heights above 360 m throughout the four seasons; such transport enhances the TKE. In spring, summer, and autumn, turbulent transport plays a very important role at altitudes of about 360 m, contributing up to 20% of the total TKE budget and is an energy inflow process that cannot be ignored. At the same time, it was found that there is a significant correlation between the rates of change of the turbulent transport term, shear generation term, and buoyancy generation term in space. Our results indicate that near the ground, buoyancy is the main generation process of TKE, contributing up to 60%. Above 570 m, the role of shear generation gradually becomes prominent, comparable to buoyancy generation. Regarding the dissipation rate, it is the only dissipation process active above 360 m during the day. At the same time, we found that there are seasonal differences in the vertical characteristics of the dissipation rate in the Shenzhen area, with maximum values near the ground in spring, summer, and autumn seasons. This is different from the results of previous studies and deserves further exploration.

While our study provides valuable insights, there are limitations to consider. The data were collected under specific weather conditions (sunny days without low clouds), which may not fully represent all atmospheric scenarios. Additionally, the dissipation rate near the ground exhibited seasonal

differences, with maximum values observed in spring, summer, and autumn, which contrasts with some previous studies and warrants further investigation.

This study not only enriches our understanding of the vertical structure of atmospheric turbulence, but also provides new observational data and theoretical support for the parameterization of turbulence energy budget terms in climate models. These findings contribute to improving the predictive ability of atmospheric dynamic processes, which is of great significance for meteorological observations, forecasting, and disaster prevention and reduction. One limitation of the current study is that the analysis does not fully account for complex weather conditions, such as precipitation and extensive cloud cover. This aspect might limit the generalizability of our findings to idealized scenarios in which clear weather conditions prevail. Future studies should examine a broader range of meteorological conditions, including significant weather disturbances, to enhance the robustness and applicability of turbulence analyses across diverse atmospheric environments.

**Data availability**

Data to generate the figures of this paper are available at https://doi.org/10.5281/zenodo.13624484 (Xian et al., 2024a).

**Author contributions**

Conceptualization: J.X. and H.Y.; methodology: J.X. and N.Z.; software: J.X.; validation: Z. Q.; formal analysis: Z.C. and N.Z.; investigation: H.Y and Z. Q.; resources: C.L. and H. R.; data curation: H. R.; writing—original draft preparation: J.X.; writing—review and editing: H.Y. and N.Z.; visualization: X.L.; supervision: H.Y. and N.Z.; project administration: H.Y. and N.Z.; and funding acquisition: H.Y. and N.Z.

**Competing interests**

The corresponding author has declared that none of the authors has any competing interests.

**Acknowledgements**

We thank Shenzhen Darsunlaser Technology Co., Ltd.

**Financial support**

This work was supported by a Shenzhen Basic Research General Project (JCYJ20240813163803006), the Key Laboratory of Urban Meteorology of the China Meteorological Administration (LUM-2024-03), the National Natural Science Foundation of China (NSFC; U2342221, and 42275065), the Key Innovation Team of China Meteorological Administration (CMA2024ZD04), and the China Postdoctoral Science Foundation (2024M751378).

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
