# Peer review of "Characteristics of Boundary Layer Turbulence Energy Budget in Shenzhen Area Based on Coherent Wind Lidar Observations"

_EGUsphere, 2025_

## Author Response (AR1)

Dear Editor and reviewers,

We would like to thank the reviewers and editor for their comments that have allowed us to further clarify some aspects of the manuscript in this revised version. Hereafter, we report reviewers' comments and our replies (*in italics*). For yours and reviewers' convenience we have put the corresponding major changes introduced in red color in the revised version of the manuscript.

**Reviewer 1:**

Profiling the atmospheric turbulence kinetic energy (TKE) is of great significance for improving our understanding of energy conversion, dissipation and material exchange in the atmospheric boundary layer, which is in turn able to affect the evolution of convection process. Due to limitations in previous detection methods, it has been challenging to visually observe the vertical structure of various budget terms of TKE, particularly the buoyancy generation term. This study provides a meaningful advancement by intuitively presenting the vertical characteristics of various TKE budget terms using high-resolution wind lidar data.

The manuscript offers a comprehensive and detailed analysis of the TKE budget in the Shenzhen area based on long-term observations from coherent wind lidar, which provides valuable insights into atmospheric boundary layer dynamics. One of the most notable and intriguing findings of this study is the discovery that the TKE tendency term transitions around 14:00 in all seasons, revealing critical insights into the diurnal variation of turbulence. The study presents important implications for atmospheric turbulence modeling and parameterization in climate models. Additionally, the analysis of seasonal variations in buoyancy and shear generation provides a more refined understanding of turbulence energy transfer processes at different altitudes.

The manuscript is well-structured, clearly written, and presents significant findings that contribute to improving our understanding of TKE budget dynamics. However, there are a few areas where minor revisions can further enhance the clarity and completeness of the study. Below are my suggested modifications:

**1)The wind profile radar can also measure the dissipation rate in the TKE budget term. Add some literature on this topic in the second paragraph of the introduction. For example, Solanki, R., et al., Elucidating the atmospheric boundary layer turbulence by combining UHF radar wind profiler and radiosonde measurements over urban area of Beijing. Urban Climate, 2022. 43,**

Response: *Thank you for the reviewer's thoughtful comments. We have carefully considered your suggestion and have incorporated the relevant literature (Solanki et al., 2022) into the introduction, as recommended. In particular, we have added a discussion on the role of wind profile radar in measuring turbulence and its limitations. The text now specifies that "Although wind profile radar can provide valuable turbulence data, its spatial and temporal resolution might be insufficient, and its ability to monitor turbulence under clear sky conditions is limited (Solanki et al., 2022)." (See lines 70 to 73)*

**2)Line 144, The assumption that pressure transport (Tp) is negligible is common in turbulence studies, but it is important to provide a clear justification for this decision. While Tp is often small in comparison to other TKE budget terms, its significance can vary depending on meteorological conditions and observational techniques. Adding supporting references on why Tp can be safely ignored in this study**

**would increase the scientific rigor of the methodology. This will also help readers unfamiliar with turbulence budget analysis better understand the reasoning behind this assumption.**

Response: *Thanks for the reviewer's professional comments. We appreciate the importance of justifying the assumption that pressure transport (Tp) is negligible, as it plays a critical role in ensuring the transparency and rigor of the methodology. As the reviewer rightly points out, while Tp is often small compared to other TKE budget terms, its relevance can indeed depend on meteorological conditions and observational techniques. However, based on previous studies, it has been consistently observed that omitting the pressure transport term generally has minimal impact on the overall turbulence analysis, particularly in typical atmospheric conditions. For example, Kaimal and Finnigan (1994) and Wyngaard (2010) suggest that Tp's contribution is often negligible in turbulent boundary layer studies, especially in well-mixed conditions. Furthermore, Pozzobon et al. (2023) confirm that in many practical applications, the pressure transport term can be safely omitted without introducing significant errors into the turbulence budget. In light of these references, we have added appropriate citations to further support this assumption and clarify its validity. As the reviewer suggests, we have added citations (Kaimal and Finnigan, 1994; Wyngaard, 2010; Pozzobon et al., 2023) in revised version. (See lines 149 to 150)*

**3)Line 323, Figure 18 provides a quantitative breakdown of the relative contributions of different TKE budget terms across various heights in the boundary layer. However, the manuscript does not clearly explain the computational method used to derive these contributions. Clarifying whether the values are obtained from normalized budget term magnitudes, fractional contributions, or other statistical methods would improve transparency.**

Response: *Thanks for the reviewer's professional comments. We understand the importance of clearly explaining the computational method used to derive the relative contributions of different TKE budget terms, as this will enhance the transparency of our analysis. In the revised manuscript, we provide a more detailed explanation of our approach. Specifically, we first categorize the TKE budget terms into two groups based on their values: positive (or zero), which we refer to as "generation" terms, and negative terms, which we classify as "dissipation" terms. After this categorization, we calculate the contribution rate of each term within its respective category by determining its proportion relative to the total value of the terms within that category. This method ensures that the contributions of each term are quantitatively expressed in a clear and logical manner. We have updated the manuscript to include this explanation for greater clarity. The text now specifies that " **We first categorized the TKE budget terms into two groups according to positive (including zero) and negative values, which were designated as generation and dissipation terms, respectively. Subsequently, we calculated the contribution rate of each budget term within its corresponding category by determining its proportion.** " (See lines 335 to 340)*

**4)The reduced number of available observational days in June and August raises questions about potential data collection biases (Table 2). Since the reliability of turbulence analysis is dependent on a continuous and representative dataset, it is crucial to clarify the cause of missing data. If the missing days are due to weather conditions (such as persistent cloud cover, heavy precipitation, or typhoon events), this should be explicitly stated.**

Response: *Thanks for the reviewer's comment. We agree that the reduced number of available observational days in June and August could raise concerns about potential biases in the data collection process, and it is important to address this issue transparently. In the revised manuscript, we have clarified the reason for the missing data. Specifically, we explain that the missing observation days are not due to any issues with the data collection process itself, but rather the result of adverse weather conditions during those months. These conditions*

*included persistent cloud cover and heavy precipitation, which significantly affected the ability to collect data. We have updated the manuscript to explicitly state this and improve transparency regarding the dataset. The text now specifies that* **"Of note, the missing observation days in June and August are due to adverse weather conditions, including continuous cloud cover and heavy precipitation, rather than data collection issues."** *(See lines 131-134)*

**5)Lines 216 and 242 respectively mention the buoyancy and shear generation terms as the reasons why TKE has the greatest impact height in autumn, but why are only buoyancy generation terms mentioned in the abstract and conclusion?**

Response: *Thank you for the reviewer's constructive comment. You are correct that both the buoyancy and shear generation terms were mentioned in lines 216 and 242 as important factors contributing to the height at which TKE has the greatest impact in autumn. However, after further quantifying the contribution rates of both, we found that the buoyancy generation term has a much more pronounced influence than the shear generation term. As shown in Figure 18(c), the contribution from buoyancy generation reaches 60%, significantly higher than the 20% contribution from shear generation. This significant difference in contribution led us to emphasize buoyancy generation as the primary driver of the high-altitude impact of autumn TKE in both the abstract and the conclusion. We have updated the manuscript to reflect this reasoning more clearly. (See lines 335 to 348)*

**6)What are the time resolution and spatial resolution of TKE budget terms? The manuscript should explicitly state the time and spatial resolution at which the TKE budget terms were derived. Resolution details are critical for interpreting turbulence measurements, as they influence how small-scale versus large-scale processes are captured. Given that the wind lidar operates at a temporal resolution of 5 seconds and a spatial resolution of 15 meters, it would be helpful to confirm whether the same resolution applies to all derived TKE terms or if additional temporal/spatial averaging was performed. Including this information in the methodology section would strengthen the manuscript's transparency and help readers better assess the scale of the analysis.**

Response: *Thank you for the reviewer's constructive comment. We agree that providing explicit details about the time and spatial resolution of the TKE budget terms is crucial for understanding the scale at which turbulence measurements are captured and for ensuring the transparency of the analysis. As the reviewer pointed out, the wind lidar operates at a temporal resolution of 5 seconds and a spatial resolution of 15 meters, but it is important to clarify whether these same resolutions were applied to all the TKE budget terms or if additional temporal/spatial averaging was performed. In the revised manuscript, we have addressed this by specifying that the TKE budget terms were derived using a time resolution of 20 minutes and a spatial resolution of 30 meters, which represents the resolution applied during the analysis. The text now specifies that* **"We obtained the horizontal wind speed (a), vertical wind speed (b), TKE (c), Et (d), Tt (e), D (f), S (g), and B (h) with a time resolution of 20 minutes and a spatial resolution of 30 m."***(See lines 150 to 152)*

**Reviewer 2:**

Due to the lack of full boundary layer turbulence detection methods in the past, it was not possible to provide the continuous vertical distribution of turbulence within the boundary layer, resulting in a certain degree of insufficient understanding of turbulence generation, transport, and dissipation processes. This study, based on high temporal and spatial resolution data from wind lidar throughout the year, inverts the vertical profiles of TKE (Turbulent

Kinetic Energy) budget terms (such as dissipation rate, shear/buoyancy generation terms, etc.), systematically revealing the diurnal and seasonal variation patterns of TKE and its budget terms, and deepening the understanding of the evolution of turbulence across the entire boundary layer. It clarifies unique phenomena in the Shenzhen area, such as the dominance of shear generation in summer and the significant buoyancy effect at high altitudes in autumn, which enhances the understanding of the boundary layer in coastal cities and provides observational evidence for climate model parameterization, offering practical application significance. The paper has a novel perspective, solid datasets, a reasonable structure, and detailed content, making it almost ready for publication. However, I believe there are still a few small issues that need attention:

**1) The lidar-based dissipation rate inversion method mentioned in the reference (Xian et al., 2025) needs to be explained in detail (e.g., is it based on inertial subrange spectrum fitting?).**

Response: *Thanks for the reviewer's professional comments. As the reviewer suggests, we have modified the texts in revised version. The text now specifies that "Based on our previous detection method, assuming $T_P$ is ignored (Kaimal and Finnigan, 1994; Wyngaard, 2010; Pozzobon et al., 2023), and obtaining dissipation rate through turbulence inertial subrange spectrum fitting, we obtained the horizontal wind speed (a), vertical wind speed (b), TKE (c), Et (d), Tt (e), D (f), S (g), and B (h) with a time resolution of 20 minutes and a spatial resolution of 30 m (Xian et al., 2025),as shown in the various panels in Figure 2." (See lines 148 to 152)*

**2)The temporal resolution of the wind lidar is 0.2 Hz/5s, but this does not necessarily mean that the turbulence kinetic energy and its budget terms have a time resolution of 5 seconds. Typically, a certain time period is required to compute the turbulence spectrum, which raises the question: is the turbulence assumed to be steady during this period? How is this ensured?**

Response: *Thanks for the reviewer's professional comments. As the reviewer suggests, we have modified the texts in revised version. The text now specifies that "**Because the time resolution was 20 minutes, ensuring steady turbulence within this period was essential. Therefore, we strictly applied a stationarity test, wherein the average variance over the entire time period was required to be similar to the average variance over shorter time intervals (Xian et al., 2025).**" (See lines 152 to 155)*

**3)The authors neglected the pressure transport term in the method. Is there any basis for neglecting this term? Please add relevant references.**

Response: *Thanks for the reviewer's professional comments. We appreciate the importance of justifying the assumption that pressure transport (Tp) is negligible, as it plays a critical role in ensuring the transparency and rigor of the methodology. As the reviewer rightly points out, while Tp is often small compared to other TKE budget terms, its relevance can indeed depend on meteorological conditions and observational techniques. However, based on previous studies, it has been consistently observed that omitting the pressure transport term generally has minimal impact on the overall turbulence analysis, particularly in typical atmospheric conditions. For example, Kaimal and Finnigan (1994) and Wyngaard (2010) suggest that Tp's contribution is often negligible in turbulent boundary layer studies, especially in well-mixed conditions. Furthermore, Pozzobon et al. (2023) confirm that in many practical applications, the pressure transport term can be safely omitted without introducing significant errors into the turbulence budget. In light of these references, we have added appropriate citations to further support this assumption and clarify its validity. As the reviewer suggests, we have added citations (Kaimal and Finnigan, 1994; Wyngaard, 2010; Pozzobon et al., 2023) in revised version.*

**4)The specific meanings of the three wind speed components in Equation 1 (u, v, w) should be provided (e.g., longitudinal direction, latitudinal direction, vertical direction).**

Response: *Thank you for the reviewer's helpful suggestion. We agree that providing a clear explanation of the three wind speed components in Equation 1 is important for improving the readability and precision of the manuscript. As recommended, we have revised the text to specify the meanings of these components. The text now specifies that "**where E represents the TKE ($m^2/s^2$), t is the time (s), and u' (longitudinal direction), v' (latitudinal direction), and w' (vertical direction) are the fluctuation values of the three-dimensional wind speed components u, v, and w, respectively, which vary with height above ground level z.**" (See line 141)*

**5)In the Introduction section, line 76-78, I agree that the author state the important of atmospheric turbulence and its impact on weather and climate change, please add relevant references. However, it is also crucial to air quality, please revise the sentence and add relevant reference (Retrieval of Boundary Layer Height and Its Influence on PM2.5 Concentration Based on Lidar Observation over Guangzhou. https://doi.org/10.46267/j.1006-8775.2021.027).**

Response: *Thank you for the reviewer's insightful suggestion. We agree that in addition to its impact on weather and climate change, atmospheric turbulence plays a crucial role in air quality, particularly in relation to particulate matter (PM2.5) concentration. As recommended, we have revised the relevant sentence in the introduction to incorporate this aspect. We also added the suggested reference to strengthen the connection between turbulence and air quality. As the reviewer suggests, we have modified the introduction and added the relevant reference in revised version. (See lines 33 to 34)*

**Reviewer 3:**

The quantitative analysis of turbulent kinetic energy (TKE) budget is a crucial support for understanding the formation mechanism of turbulence. This paper, using coherent Doppler lidar observation data, visually presents the vertical structure and seasonal variation characteristics of TKE budget in the boundary layer of Shenzhen area. The research method of this paper has obvious innovation, and the conclusions obtained also have scientific significance. The study reveals the vertical distribution characteristics of TKE in different seasons in the coastal area of South China (for example, the TKE in the lower layer is the strongest in summer), and clarifies the relative contributions of buoyancy generation and shear generation. The features of the time evolution analysis (for example, the transformation of the trend term of TKE around 14:00) and the explanation of the mechanism (buoyancy generation dominates) are logically clear. This research provides new evidence for the formation mechanism of the boundary layer turbulence in subtropical coastal cities and theoretical support and data support for the parameterization scheme of the boundary layer turbulence process. Although this research is structurally rigorous and scientifically strong, some technical details and methodological explanations still need to be provided to enhance the completeness and universality of the paper.

**1) (Line 140): The symbol $\theta$ is referenced in the TKE budget equation but lacks explicit definition. Given the critical role of thermal stratification in buoyancy-driven turbulence, the potential temperature ($\theta$)should be explicitly defined to avoid ambiguity.**

Response: *Thank you for the reviewer's valuable comment. We appreciate the importance of clearly defining all symbols used in the manuscript, especially for key variables like the potential temperature (θ). As suggested, we have updated the manuscript to explicitly define the symbol θ to avoid any ambiguity. Specifically, θm represents the mean potential temperature, while θ' denotes the fluctuation in potential temperature. This clarification ensures that readers will have a clear understanding of the role of thermal stratification in buoyancy-driven turbulence. The revised text now reads: "**θm represents the mean potential temperature, and θ' denotes the fluctuation of the potential temperature.**" (See lines 144 to 145)*

**2)Vertical Coordinate Clarification (Line 138): The vertical coordinate z is ambiguously described as "height." Please specify whether z represents altitude above mean sea level (AMSL) or height above ground level (AGL) in the methodology section.**

Response: *Thank you for the reviewer's helpful comment. We agree that providing a clear definition of the vertical coordinate is essential for avoiding any ambiguity. In the revised manuscript, we have clarified that the vertical coordinate z refers to the height above ground level (AGL), not altitude above mean sea level (AMSL). This distinction is important for ensuring that the methodology is clearly understood. The text now specifies that "where E represents the TKE ($m^2/s^2$), t is the time (s), and u' (longitudinal direction), v' (latitudinal direction), and w' (vertical direction) are the fluctuation values of the three-dimensional wind speed components u, v, and w, respectively, which vary **with height above ground level z**." (See lines 140 to 145)*

**3)Pressure Transport Term (Line 144): The manuscript omits the pressure transport term (Tp) in the TKE budget analysis without theoretical or empirical justification. Cite peer-reviewed studies (e.g., Zhou et al., 1985; Nilsson et al., 2016a) that validate the negligible contribution of Tp in similar boundary layer regimes to strengthen methodological credibility.**

Response: *Thanks for the reviewer's professional comments. We appreciate the importance of justifying the assumption that pressure transport (Tp) is negligible, as it plays a critical role in ensuring the transparency and rigor of the methodology. As the reviewer rightly points out, while Tp is often small compared to other TKE budget terms, its relevance can indeed depend on meteorological conditions and observational techniques. However, based on previous studies, it has been consistently observed that omitting the pressure transport term generally has minimal impact on the overall turbulence analysis, particularly in typical atmospheric conditions. For example, Kaimal and Finnigan (1994) and Wyngaard (2010) suggest that Tp's contribution is often negligible in turbulent boundary layer studies, especially in well-mixed conditions. Furthermore, Pozzobon et al. (2023) confirm that in many practical applications, the pressure transport term can be safely omitted without introducing significant errors into the turbulence budget. In light of these references, we have added appropriate citations to further support this assumption and clarify its validity. As the reviewer suggests, we have added citations (Kaimal and Finnigan, 1994; Wyngaard, 2010; Pozzobon et al., 2023) in revised version.*

**4)Data Generalizability Limitations: the exclusion of complex weather conditions (e.g., precipitation, cloud cover) limits the applicability of findings to idealized scenarios. Explicitly acknowledge this limitation in the Conclusions section, emphasizing the need for future studies under diverse meteorological conditions.**

Response: *As the reviewer suggests, we have modified the texts in revised version. The text now specifies that "**One limitation of the current study is that the analysis does not fully account for complex weather conditions, such as precipitation and extensive cloud cover. This aspect might limit the generalizability of our findings to**

*idealized scenarios in which clear weather conditions prevail. Future studies should examine a broader range of meteorological conditions, including significant weather disturbances, to enhance the robustness and applicability of turbulence analyses across diverse atmospheric environments.*"(See lines 382-387)

**5)Temporal Reference in Figures: Figures 3, 5, 7, 9, and 10 display diurnal cycles without specifying the time zone. Label all temporal axes as "Local Time (UTC+8)" to align with Shenzhen's geographic context to avoid misunderstanding.**

Response: *Thank you for the reviewer's thoughtful comment. We agree that specifying the time zone is crucial for clarity, particularly when displaying diurnal cycles, to avoid any confusion. In the revised manuscript, we have addressed this by explicitly stating the time zone for all temporal references. Specifically, we have added that all times mentioned in this study are in local time (UTC+8), corresponding to Shenzhen's geographic context. The text now specifies that "Figure 4(b) shows mean TKE profiles at 13:00 for each season (**all times mentioned in this study are in local time**)."(See lines 173 to 174)*

We thank the reviewers again for their helpful suggestions, which have helped us to improve this manuscript quite much.

We also thank the Editor again for his helpful suggestions.

On behalf of all authors,
Sincerely,
Honglong Yang

Shenzhen National Climate Observatory
Meteorological Bureau of Shenzhen Municipality
518000 Shenzhen, China
E-mail: yanghl01@163.com

---

## Author Response (AR2)

**EGUSPHERE-2025-157- Response Letter**

Dear Editor and reviewers,

We would like to thank the reviewers and editor for their comments that have allowed us to further clarify some aspects of the manuscript in this revised version. Hereafter, we report reviewers' comments and our replies (*in italics*). For yours and reviewers' convenience we have put the corresponding major changes introduced in red color in the revised version of the manuscript.

**Reviewer 2:**

 **The authors have provided a very good response to my comments. I suggest to accept the manuscript in its current form. But a small issue needs to be noted:**

**Line 20: "the dissipation rate term is t is the" seems to be a typo.**

Response: *Thanks for the reviewer's professional comments. As the reviewer suggests, we have modified the texts in revised version. (See line 20)*

We thank the reviewers again for their helpful suggestions, which have helped us to improve this manuscript quite much.

We also thank the Editor again for his helpful suggestions.

On behalf of all authors,
Sincerely,
Honglong Yang

Shenzhen National Climate Observatory
Meteorological Bureau of Shenzhen Municipality
518000 Shenzhen, China
E-mail:  yanghl01@163.com